# Making sense of oxygen sensing

Peter J Ratcliffe [1,2,3]✉ & Thomas P Keeley [1,2,4]✉

**Homeostatic control of cellular oxygen availability is a crucial feature of all eukaryotic life, and central to this process is the ability to sense oxygen across a broad range of concentrations and time scales. Much of our understanding of the molecular mechanisms underpinning oxygen sensing has been obtained using cell culture models, yet the biophysical properties of oxygen combined with the complex nature of cellular $O_2$ consumption can make the interpretation of such data difficult. In this commentary, we have outlined some of the main problems encountered in measuring and manipulating cell monolayer oxygenation in vitro, and contextualised them using both historical and contemporary examples.**

See also: Metabolism Methods Commentary Series

Oxygen homeostasis in multicellular life is highly complicated, involving cellular, tissue, and whole-organismal feedback loops tightly integrated to regulate cellular oxygen levels within a narrow range. Unlike many other physiological parameters, this setpoint will differ substantially between cell type and tissue localisation. Furthermore, substantial yet often transient deviations in supply or demand have forced the evolution of oxygen homeostatic loops acting across a wide range of time windows. Rapid cardiorespiratory reflex responses operate within seconds to normalise arterial oxygen content (Bishop and Ratcliffe, 2025), whilst the hypoxia-inducible factor/prolyl hydroxylase (HIF–PHD) pathway acts to promote transcriptional adaption to chronic oxygen deprivation (Missiaen et al, 2023). Arguably, the control of mammalian cellular oxygen levels is the archetypal example of physiological homeostasis. Much of this understanding comes from our interpretation of, and ability to manipulate, oxygen concentration in vitro. Indeed, separating the effects of oxygen depletion (pure hypoxia) from confounding factors that also change as a result of hypoxia, such as cellular proliferation and metabolism, together with the development of technology to accurately control ambient oxygen levels, were key to unravelling the mechanism of widespread oxygen sensing through the HIF–PHD pathway. In this commentary, we will give a brief overview of the problems associated with studying oxygen homeostasis in vitro, then highlight both historical and contemporary examples where the acknowledgement of these problems, and innovative approaches to overcome them, have furthered our understanding of the underlying physiology.

## The problem with oxygen in vitro

The complexity of the mammalian cardiorespiratory system reflects the problem of distribution in an oxygen-rich atmosphere. It is therefore unsurprising that the practice of culturing cells under simple monolayer conditions falls short in recapitulating this. The details of this problem have been extensively described elsewhere (Keeley and Mann, 2019; Place et al, 2017), and hence we summarise these considerations in two main points: (i) intracellular oxygen levels are nearly always inferred by reference to alterations in ambient $O_2$, and seldom measured directly, and (ii) the combination of a static layer of medium, relatively impermeable plastic cultureware, and monolayer oxygen consumption can create significant gradients that pose a barrier to adequate oxygen supply. Diffusion kinetics in monolayer culture can be simplified if we assume that medium/plasticware composition (i.e., diffusivity and solubility constants) and volume (hence liquid depth, $d$), remain relatively constant between laboratories. With these assumptions, monolayer oxygenation is principally dependent on ambient oxygen level ($Ps$) and monolayer oxygen consumption, the latter a function of single cell oxygen consumption rate ($V$) and monolayer density ($P$) (Fig. 1). Accordingly, variations in any of these three parameters can radically influence overall monolayer oxygenation, with profound consequences for the interpretation of experimental data, as the rest of this article will explore.

## Cell density and the demonstration of cellular oxygen sensing

The origins of the discovery of widespread oxygen sensing through the PHD-HIF pathway can be found, in part, in the desire to understand the molecular basis of hypoxaemia-dependent erythropoiesis (Ratcliffe, 2022). A major breakthrough came from the discovery that certain hepatocyte cell lines, HepG2 and Hep3B, demonstrated oxygen-dependent EPO mRNA induction with concomitant secretion of EPO protein (Goldberg et al, 1987). Two extremely important observations were made in this report: Firstly, that EPO production increased in proportion to cell density when cultured at atmospheric oxygen levels, and secondly, that hypoxic induction was only observed in low-density cultures. This is likely because HepG2 and Hep3B lines will readily form highly dense, clustered monolayers in culture and are severely oxygen-limited even when cultured under ambient atmospheric oxygen concentrations (Bhatia et al, 1996; Metzen et al, 1995; Ng et al, 2014; Wolff et al, 1993). In line with this, EPO production can be stimulated in other cell lines but usually only when exposed to very low ambient oxygen levels (0.2% $O_2$) (Stolze et al, 2002). Note, true anoxia is

[1]Target Discovery Institute, Nuffield Department of Medicine, University of Oxford, Oxford OX3 7FZ, UK. [2]Ludwig Institute for Cancer Research, Nuffield Department of Medicine, University of Oxford, Oxford OX3 7FZ, UK. [3]The Francis Crick Institute, 1 Midland Road, London NW1 1AT, UK. [4]Department of Physiology, Anatomy and Genetics, University of Oxford, Oxford OX1 3PT, UK. ✉E-mail: peter.ratcliffe@ndm.ox.ac.uk; thomas.keeley@dpag.ox.ac.uk
https://doi.org/10.1038/s44318-025-00513-1 | Published online: 1 August 2025

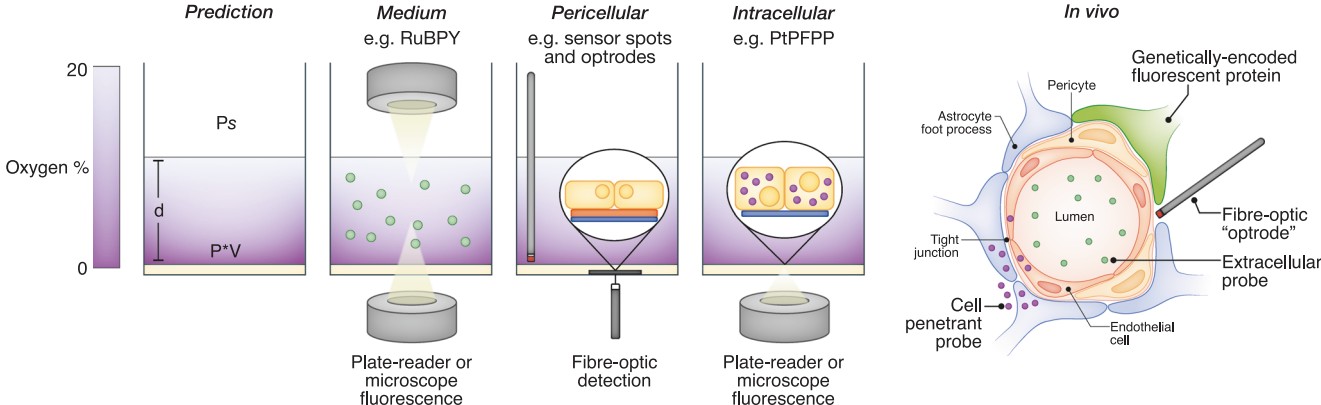

**Figure 1. Measuring oxygen distribution.**

The formation of gradients in oxygen concentration in a static monolayer of medium above a monolayer of cultured cells. Ambient oxygen levels in the headspace ($P_s$), depth/volume of medium used (d), and monolayer oxygen consumption (derived from a combination of single cell oxygen consumption rate (V) and cell density(P)) are major determinants of this gradient. These variables can be used to mathematically model oxygen distribution in vitro (prediction), which can also be experimentally determined using an array of fluorescent/phosphorescent indicators to accurately map dissolved, pericellular, and/or intracellular oxygen concentrations in cultured cells, and oxygen distribution in vivo. RuBPY Tris(bipyridine)ruthenium(II), PtPFPP Pt(II)-tetrakis(pentafluorophenyl)porphyrin. Far right: brain capillary illustration adapted from (Goldstein and Betz, 1986).

quite difficult to achieve rapidly in culture, since polystyrene culture vessels act as a sink of oxygen from which desorption will occur when exposed to a gradient (Stevens, 1992). The careful description of conditions required for EPO secretion opened the field to a robust dissection of the molecular mechanisms underpinning EPO mRNA regulation, and subsequently to the discovery of widespread oxygen sensing through the HIF–PHD pathway.

## Is oxygen consumption a more widespread problem?

The example above relates to a highly regulated, directly oxygen-sensitive gene, which is only expressed in a small subset of specialised cell types. Yet oxygen is important for many cellular functions, most of which exhibit apparently lower sensitivity to oxygen within the physiological range. A prime example of this is oxidative phosphorylation, which relies on oxygen as a terminal electron acceptor but is not thought to be limited by oxygen concentration until the cell approaches anoxia (Gnaiger, 2003). It is widely accepted that the combination of culture medium with excessive amounts of glucose and other metabolites, and the Warburg effect, lead to most immortalised cell lines utilising glycolysis as their primary energy source (Cantor et al, 2017). However, this assumption was recently challenged by Tan and

colleagues(Tan et al, 2024), who proposed that the conditions in which many cell types are cultured lead to a depletion in oxygen availability to such an extent that oxidative phosphorylation does become rate-limited. By simply reducing the volume of medium in which cells were cultured, the authors were able to increase pericellular oxygen availability substantially (see Fig. 1), resulting in increased oxygen consumption as cells reverted back towards oxidative phosphorylation. Similar metabolic reprogramming following restoration of adequate oxygen supply has been reported previously, in renal epithelia using a roller bottle (adherent monolayer, stirred media) model (Gstraunthaler et al, 1999), in hepatocytes using a stirred-suspension model (Gilglioni et al, 2018) and in breast cancer cell lines by using mathematical modelling (Rogers et al, 2024). While this may not be a newly described phenomenon, it is the accuracy in defining the issue combined with the simplicity and technical accessibility of the solution proposed by Tan and colleagues, that we considered worth highlighting specifically.

## Trouble at the other extreme

Many cell types either do not consume enough oxygen, or grow at insufficient density to create an oxygen gradient that limits supply, presenting the opposite issue in which supply outweighs demand and

resulting in exposure to supraphysiological levels of oxygen. This problem has generally been addressed by reducing ambient oxygen levels until a pre-established desirable level of monolayer oxygenation is achieved, usually defined with respect to the normal oxygen concentration in the parent tissue (Keeley and Mann, 2019). Given the limitless supply in the atmosphere, it is technically more demanding to reduce exposure to oxygen than it is to increase oxygen delivery, requiring specialist hypoxia workstations. Nonetheless, greater effort is being made to define and recapitulate physiological normoxia in vitro. Endothelial cell culture exemplifies this problem, since these cells preferentially utilise glycolysis for ATP generation both in vivo and in culture (Culic et al, 1997), consuming much lower amounts of oxygen than the highly metabolically active cell lines discussed above. Consistent with this, oxygen gradients between the atmosphere and pericellular (Abaci et al, 2010) and intracellular (Chapple et al, 2016) environment in endothelial monolayers are very slight, if detectable at all. With blood oxygen levels equivalent to 3–10% (Keeley and Mann, 2019), this means these cultures are exposed to oxygen concentrations far above their physiological level when cultured under standard atmospheric (circa 18–20%) conditions. This has been rectified through the use of hypoxia workstations set to 5% oxygen (and 5% $CO_2$), permitting cell culture under

physiologically normoxic conditions. This can result in important phenotypic changes; for instance, human umbilical vein endothelial cells cultured under these conditions consistently demonstrate phenotypes distinct from paired cultures maintained at either hyperoxia (ambient) or hypoxia (1% $O_2$) (Abaci et al, 2010; Keeley et al, 2017).

## Deviations from the steady state

The examples given above represent situations in which a steady-state equilibrium is sought between the baseline cellular phenotype (oxygen consumption, monolayer density) and the ambient atmospheric supply of oxygen. The final scenario we would like to explore is one in which experimental interventions induce a change in cellular oxygen consumption to an extent that has a direct effect on oxygen availability, with consequent modulation of cellular oxygen-sensitive processes. If such an effect is unintentional or unknown, it can lead to the erroneous conclusion that the intervention has a direct effect on the oxygen-sensitive process itself. An example of this is the interaction between nitric oxide (NO) and the PHD-HIF pathway, for which numerous mechanisms have been suggested (Berchner-Pfannschmidt et al, 2010), but have largely focused on direct post-translation modifications induced by high concentrations of exogenous NO applied under hyperoxic conditions. At physiological concentrations (of both gases), however, a major action of NO is to inhibit oxygen consumption by cytochrome C oxidase through competitive antagonism with oxygen (Brown and Cooper, 1994). This led to the demonstration by Hagen and colleagues (Hagen et al, 2003) that the ability of NO to reduce the accumulation of HIF-1α protein under hypoxia was through an increase in cytosolic oxygen availability and not due to a direct effect on degradation pathways. Another example of this problem is the activation of inflammatory cells in vitro, which leads to a large increase in the activity of oxygen-consuming processes such as the respiratory burst (Colgan and Taylor, 2010). In an elegant dissection of this interaction, Campbell and colleagues show that activation of peripheral mononucleocytes using N-Formylmethionine-leucyl-phenylalanine induced a hypoxic gene signature in co-cultured epithelial cells (Campbell et al, 2014). Through direct measurements of pericellular oxygen concentrations of the

co-cultured epithelia, it was established that neutrophil activation led to oxygen depletion in the media in a manner dependent on cell density and NAD(P)H oxidase activity. In both of these examples, an appreciation of the effect on oxygen consumption and effort spent to assess this possibility experimentally avoided the incorrect conclusion that either intervention had a direct (oxygen-independent) effect on HIF pathway activity.

## The future is phosphorescent

The advances in technology aimed at measuring peri/intracellular oxygen levels in recent years have highlighted the problems with assuming equivalency (or even linearity) between ambient and monolayer oxygen concentrations. Direct physical measurements using oxygen-sensitive electrodes are possible but suffer from difficulty in obtaining accurate proximity to intact cells, the potential to generate trauma, or confounding through the consumption of oxygen. Detailed reviews of all available technology for determining oxygen levels in culture can be found elsewhere (Keeley and Mann, 2019; Place et al, 2017). Here we would like to briefly discuss the ruthenium/platinum protoporphyrin-based measurements and their three forms of application, which together appear particularly suitable for spatially defined measurements in biological systems. First described in 1987 (Vanderkooi et al, 1987), these compounds exhibit long phosphorescence decay lifetimes that are reversibly modulated by oxygen binding. In conjunction with a substantial separation between excitation and emission maxima, these physicochemical properties make for highly accurate readings with extremely high signal-to-noise values. Moreover, the reversible nature of the signal allows for real-time and repeated measurements.

The compounds are versatile and can be constructed as: (i) hydrophilic, cell-impermeable complexes to monitor dissolved oxygen in biological fluids (Blaszczak et al, 2021; Torres Filho and Intaglietta, 1993), (ii) immobilised sensor films on, or adjacent to which cells can be grown, to report pericellular oxygen levels (Campbell et al, 2014; Tan et al, 2024), and (iii) conjugated to carrier polymers to produce cell-permeable probes that can directly monitor intracellular oxygen levels (Chapple et al, 2016; Fercher et al, 2011; Zhdanov

et al, 2008). Unfortunately, the latter exhibit a heterogenous distribution within the cell (Fercher et al, 2011), currently precluding the assessment of intracellular oxygen gradients. Notably, improvements in the oxygen sensitivity of bio-reductive dyes (Wallabregue et al, 2023) and the utility of genetically encoded oxygen-sensitive fluorescent reporter proteins(Erapaneedi et al, 2016) may allow for greater subcellular resolution in the future.

The minimally invasive nature and lifetime-based quantification of phosphorescent oxygen probes have also made them ideal tools to measure oxygen levels in vivo. In combination with 2-photon microscopy, this has been used to great effect in mapping $O_2$ distribution within cerebral capillaries (Lecoq et al, 2011) using non-cell penetrant platinum-porphyrin-PEG conjugates. Cell penetrant equivalents have been used to map $O_2$ distribution in the vascular and extravascular bone marrow compartments (Spencer et al, 2014), demarcated using fluorescent dextran to mark the blood vessels. More recently, a genetically encoded oxygen reporter protein utilising the oxygen-dependent, FRET-amplified bioluminescence of green enhanced nano-lantern has been used to monitor cerebral tissue $O_2$ levels (Beinlich et al, 2024). Both these technologies offer the exciting opportunity to measure changes in oxygen distribution under more physiologically relevant conditions, although the demanding and expensive equipment required to use them is likely to limit widespread adoption.

## Conclusion

Oxygen homoeostasis is a critical physiological challenge and requires the sensing of oxygen levels coupled to feedback systems that adjust supply or demand. Such controls may be achieved through the direct sensing of oxygen or indirectly through the sensing of metabolites that are affected by oxygen availability. Direct oxygen sensing is of particular interest because, potentially, it regulates physiological pathways that are specific to oxygen homeostasis and provides routes to specific therapeutic modulation. The use of hypoxic tissue culture, in which low oxygen can be applied with limited confounding effects from the compromise of metabolism and cellular energetics that dominate ischaemic stresses, was critical in enabling the discovery of a widespread system of direct oxygen sensing operating

in animal cells. Moreover, oxygen levels can be controlled in tissue culture in a graded fashion that represents the physiology in a meaningful way, thus facilitating the extrapolation of results to integrated systems. Nevertheless, the common practice of inferring oxygen levels from the ambient tissue culture atmospheric concentration entrains uncertainties and risks the inadvertent exposure of cells to hypoxia or hyperoxia. Advances in the direct measurements of pericellular oxygen or even intracellular oxygen concentration hold the potential to mitigate these problems. Meanwhile, awareness of the risks in inferring cellular oxygen levels from atmospheric concentrations and the appreciation of means of mitigating such errors are important.

## Key considerations

- Oxygen concentration at the level of the monolayer is strongly influenced by cellular density, oxygen consumption rate, ambient oxygen levels, and medium volume.
- Accordingly, the availability of oxygen can vary considerably between different cell lines and culture conditions, suggesting the need for standardisation in cell culture practices
- Experimental interventions that impact on cell oxygen consumption rates can produce substantial and rapid changes in cellular oxygen concentration, with downstream effects on oxygen-sensitive pathways
- Measuring oxygen levels in cell culture can inform the design of more physiologically relevant models, as well as prevent the misinterpretation of experimental findings related to oxygen sensing processes, and beyond.

## Peer review information

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

## Acknowledgements

PJR and TPK are supported by the Wellcome Trust (301530/Z/23/Z), and TPK is also supported by the British Heart Foundation (FS/IBSRF/24/25194). PJR is also supported by the Francis Crick Institute, which receives funding from Cancer Research UK, (FC001501, the UK Medical Research Council (FC001501), and the Wellcome Trust (FC001501). PJR is also supported by the Ludwig Institute for Cancer Research, Oxford, where he is a Distinguished Scholar.

## Author contributions

**Peter J Ratcliffe**: Conceptualisation; Writing—review and editing. **Thomas P Keeley**: Conceptualisation; Writing—original draft; Writing—review and editing.

## Disclosure and competing interests statement

PJR is a non-executive director of Immunocore Plc. and is a member of the *EMBO Journal* editorial advisory board.

