## [Peer Review File · The EMBO Journal]

Making sense of oxygen sensing

Peter Ratcliffe and Thomas Keeley

Corresponding authors: Thomas Keeley (thomas.keeley@dpag.ox.ac.uk), Peter Ratcliffe (peter.ratcliffe@ndm.ox.ac.uk)

Review Timeline:

Submission Date:	15th May 25
Editorial Decision:	5th Jun 25
Revision Received:	23rd Jun 25
Accepted:	1st Jul 25

Editor: Daniel Klimmeck

Transaction Report:

Dear Thomas, dear Peter,

Thank you again for sending us your commentary article for the metabolism advice series. As mentioned, we have asked two dedicated hypoxia experts to assess your manuscript, and in the meantime got feedback from both of them, which I enclose below.

As you will see, the colleagues much appreciate the perspective piece and find it timely and worth publishing. They also provide constructive feedback on how to further improve it by advancing the discussion by including aspects on in vivo applications, and utility of sensors available. They also suggest enhancements of how to represent key concepts covered in the figure.

Please note that in our view referee #2's point on experimental standards is well taken, however we reason that formulating such standards and potential consensus views for the field is likely beyond the framework of this commentary.

With respect to the artwork, please consider adjustments and let me know about your views about the referee's point. In any case, a preliminary next version of the sketch will be sufficient as a basis for final rendering by our scientific illustration team.

I hope you will find the comments helpful. I am sure that an amended version incorporating the suggestions made by the referees will be highly noted and appreciated. I would thus like to invite you to submit such a revised version using the link enclosed below.

Please let me know in case I can be of any help with this.

with
Best wishes to Oxford/London,

Daniel

Daniel Klimmeck, PhD
Senior Editor
The EMBO Journal

Referee #1:

The Perspective by Ratcliffe and Keeley, entitled "Making Sense of Oxygen Sensing", offers an improved conceptual framework for furthering our mechanistic insight into how homeostatic control of cellular oxygen availability is assessed. While much of our detailed understanding of the molecular mechanisms underpinning oxygen sensing has been derived from cell culture models, the biophysical properties of oxygen-combined with cellular consumption-can make experimental interpretation challenging. In this commentary, the authors outline several key issues involved in measuring and modulating oxygen levels in monolayer cell cultures in vitro. They also provide both historical context and contemporary examples to support their discussion. I recommend this article for publication, pending consideration of the following points:

1. In Vivo Relevance of Oxygen-Sensing Probes:

The authors provide a detailed discussion on how oxygen gradients in monolayer cultures are influenced by factors such as media volume and cellular oxygen consumption. It would strengthen the article to include a section on how some of the more recent and promising oxygen-sensing probes could be adapted for in vivo applications. While monolayer systems have revealed a great deal about core oxygen-sensing pathways, they do not fully replicate the complexity of tissue environments, particularly

in pathological and physiological contexts.

2. Application of ruthenium/platinum-protoporphyrin Based Sensors:

The authors rightly highlight the discrepancies between ambient oxygen levels and intracellular oxygen concentrations, and they note the utility of ruthenium- and platinum-based probes (e.g., protoporphyrin derivatives) for more accurate measurements. It would be valuable for the reader to understand how these technologies might be applied in 3D culture systems or in vivo models, where oxygen gradients are even more physiologically relevant.

3. Minor Editorial Suggestion:

In the section titled "Is Oxygen Consumption a More Widespread Problem?", the second sentence reads:

"That which relies of oxygen as a terminal electron..."

This should be corrected to:

"That which relies on oxygen as a terminal electron acceptor..."

Referee #2:

This commentary by Ratcliffe and Keeley explores the complexities and pitfalls of studying oxygen sensing in vitro, emphasizing that oxygen availability at the cellular level is highly variable and influenced by factors such as cell density, culture medium depth, and oxygen consumption rates. Traditional monolayer cell culture often fails to replicate physiological oxygen conditions, leading to misleading conclusions about hypoxia-related processes. The authors review both historical and recent studies that have uncovered how experimental conditions can unintentionally alter oxygen availability, thereby affecting the hypoxia-inducible factor (HIF) pathway and other oxygen-sensitive mechanisms. They highlight the growing importance of direct measurements of pericellular and intracellular oxygen using phosphorescent and fluorescent probes to build more accurate, physiologically relevant in vitro models. Importantly, this analysis reinforces recent findings by Tan et al. (EMBO J 2024), who demonstrated oxygen limitation as a key modulator of metabolic phenotype in culture, as well as the commentary by Mazure (EMBO J 2024), which emphasizes the need to rethink how oxygen is monitored and controlled in cell culture experiments. Overall, the commentary is pedagogical, forward-thinking, and reinforces the importance of re-evaluating experimental design in cellular oxygen research.

Although the commentary effectively highlights the importance of direct oxygen measurements in vitro and outlines key technologies—such as phosphorescent probes based on ruthenium or platinum, pericellular sensor films, and intracellular oxygen-sensitive dyes—it would have been valuable to include a visual summary or schematic. A diagram illustrating the spatial scale of these different measurement approaches (media, pericellular space, intracellular compartments) and their respective advantages or limitations would greatly help readers unfamiliar with these tools to appreciate their potential applications and technical considerations. In contrast, Figure 1, while conceptually relevant, adds limited value given that these biophysical principles have already been extensively developed and illustrated in the work of Tan et al. (EMBO J 2024). A figure dedicated to oxygen-sensing technologies would have been more informative and aligned with the article's forward-looking perspective. Given that the authors' recognition for their seminal contributions to the field of oxygen sensing, this commentary has the authority to do more than highlight technical challenges. It would be highly valuable if the authors went beyond general recommendations to propose more structured frameworks or experimental standards for oxygen control in vitro. Their expertise uniquely positions them to help define practical guidelines that could drive greater consistency and physiological relevance in hypoxia-related research. Such contributions would be particularly impactful in shaping future practices across the field.

The authors addressed the remaining editorial issues.

Dear Peter, dear Thomas,

Thank you for sending us the updated final version of the commentary article.

I am pleased to inform you that your manuscript has been accepted for publication in the EMBO Journal.

Your manuscript will be processed for publication by EMBO Press. It will be copy edited and you will receive page proofs prior to publication. Please note that you will be contacted by Springer Nature Author Services to complete licensing and payment information. Please note also that as this is invited front-half content, OA charges applicable to this article will be covered. Please use the following token -XXXXXXXXXXXX- when entering the licensing process.

Further, our scientific graphics illustrator Luk Cox is currently converting the commentary figure into journal style. He will contact you shortly on the proof stage image for your input.

If you have any questions, please do not hesitate to contact me.

Thank you again for your kind contribution to The EMBO Journal, which is much appreciated.

with
Best regards,

Daniel

Daniel Klimmeck, PhD
Senior Editor
The EMBO Journal